# Evaluating Biofilm Inhibitory Potential in Fish Pathogen, *Aeromonas hydrophila* by Agricultural Waste Extracts and Assessment of Aerolysin Inhibitors Using *In Silico* Approach

**DOI:** 10.3390/antibiotics12050891

**Published:** 2023-05-11

**Authors:** Manikandan Arumugam, Dinesh Babu Manikandan, Sathish Kumar Marimuthu, Govarthanan Muthusamy, Zulhisyam Abdul Kari, Guillermo Téllez-Isaías, Thirumurugan Ramasamy

**Affiliations:** 1Laboratory of Aquabiotics/Nanoscience, Department of Animal Science, School of Life Sciences, Bharathidasan University, Tiruchirappalli 620024, India; 2Department of Pharmaceutical Technology, University College of Engineering, Bharathidasan Institute of Technology (BIT) Campus, Anna University, Tiruchirappalli 620024, India; 3Department of Environmental Engineering, Kyungpook National University, Daegu 41566, Republic of Korea; 4Department of Agricultural Sciences, Faculty of Agro‐Based Industry, Jeli Campus, Universiti Malaysia Kelantan, Jeli 17600, Malaysia; 5Advanced Livestock and Aquaculture Research Group, Faculty of Agro-Based Industry, Jeli Campus, Universiti Malaysia Kelantan, Jeli 17600, Malaysia; 6Department of Poultry Science, University of Arkansas, Fayetteville, AR 72701, USA

**Keywords:** aerolysin, agri-waste, antimicrobial metabolites, molecular docking and dynamics, quorum sensing

## Abstract

*Aeromonas hydrophila*, an opportunistic bacteria, causes several devastating diseases in humans and animals, particularly aquatic species. Antibiotics have been constrained by the rise of antibiotic resistance caused by drug overuse. Therefore, new strategies are required to prevent appropriate antibiotic inability from antibiotic-resistant strains. Aerolysin is essential for *A. hydrophila* pathogenesis and has been proposed as a potential target for inventing drugs with anti-virulence properties. It is a unique method of disease prevention in fish to block the quorum-sensing mechanism of *A. hydrophila*. In SEM analysis, the crude solvent extracts of both groundnut shells and black gram pods exhibited a reduction of aerolysin formation and biofilm matrix formation by blocking the QS in *A. hydrophila*. Morphological changes were identified in the extracts treated bacterial cells. Furthermore, in previous studies, 34 ligands were identified with potential antibacterial metabolites from agricultural wastes, groundnut shells, and black gram pods using a literature survey. Twelve potent metabolites showed interactions between aerolysin and metabolites during molecular docking analysis, in that H-Pyran-4-one-2,3 dihydro-3,5 dihydroxy-6-methyl (−5.3 kcal/mol) and 2-Hexyldecanoic acid (−5.2 kcal/mol) showed promising results with potential hydrogen bond interactions with aerolysin. These metabolites showed a better binding affinity with aerolysin for 100 ns in molecular simulation dynamics. These findings point to a novel strategy for developing drugs using metabolites from agricultural wastes that may be feasible pharmacological solutions for treating *A. hydrophila* infections for the betterment of aquaculture.

## 1. Introduction

Although aquaculture is one of the industries producing food with the greatest growth rate, bacterial fish infections result in large output losses every year [1]. The disease is a significant negative socioeconomic consequence for those dependent on aquaculture and a major barrier to aquaculture operations [2]. With the growth of aquaculture activities, stress conditions also increase, encouraging the frequent incidence and development of pathogens [3,4]. Furthermore, approximately 10 to 50% of output loss is brought on by epizootics, which severely hampers the efforts to increase productivity [5,6]. *Aeromonas hydrophila* is a freshwater chemoorganoheterotrophic, facultatively anaerobic, gram-negative pathogenic bacteria that mostly affects fish, mammals, birds, amphibians, and reptiles. It causes infections such as gastroenteritis, necrotizing fasciitis, and septicemia in the species mentioned above [7,8]. Since the condition is driven by several virulence factors, including cytotoxins, adhesions, hemolysins, proteases, lipases, and biofilm development, *A. hydrophila* is multifaceted in its pathogenicity [9,10].

Aerolysin causes symptoms of many sorts of infections, including hemorrhagic and ulcerative lesions on the skin and other organs [11,12]. Aerolysin has numerous effects such as hemolytic, enterotoxin, and cytotoxic activities [13,14,15]. Aerolysin can enter the target cell’s membrane after producing seven oligomeric subunits with a transmembrane pore [16,17,18]. The channel pore breaches the cellular membrane permeability barrier, resulting in cell death [19]. One of the main virulence factors in developing fish disorders linked to *A. hydrophila* is the gene aerolysin, a destructive pore-forming enterotoxin [20]. Tetramycin and romet-30 are the most widely and frequently used antimicrobial drugs against *A. hydrophila* contamination in freshwater aquaculture [2]. The key pathogenic factors assisted by *Aeromonas* sp. are surface polysaccharides, extracellular proteins, iron-binding systems, and exotoxins, which are crucial in the pathogenic mechanisms. These virulence factors have antibiotic resistance that might develop in aquaculture systems; nevertheless, these antimicrobial drugs are used indiscriminately [21,22,23]. These antibiotics are mostly administered directly to the aquaculture system by feed or submersion. According to the scientific literature, between 70 and 80% of prescribed antibiotics eventually enter water [24,25,26,27]. 

As an outcome, the aquaculture systems have been designated as “biological hubs” for bacterial transduction, conjugation, and transformation of antibiotic-resistant genes [28,29]. Therefore, as our reliance on aquaculture increases, it is vital to investigate appropriate antibiotic substitutes that feedstuffs may deliver, lower the risk of antimicrobial resistance emerging, and increase the fish immune system [30,31,32]. Various approaches have been recently proposed to combat the rise of antibiotic resistance, including the use of plant metabolites to improve and stimulate the fish’s immune characteristics in aquaculture [33]. Plants and their secondary metabolites have a wide range of activities, which raises the possibility that they could be used as antimicrobial agents. In particular, the main natural plant species come under the *Apiaceae*, *Anacardiaceae*, *Burseraceae*, *Cupressaceae*, *Dracenaceae*, *Euphorbiaceae*, *Fabaceae*, *Palmaceae*, and *Pinaceae* families of plants [34,35]. Tannins are the phenolic polymers in all plants that tend to inhibit the bacterial matrices, outer membranes, and protein transport in bacterial cells and may prevent several hydrolytic enzymes such as α-amylase which is essential for cell growth [36]. Several plant natural products or secondary metabolites have been shown to positively impact disease virulence factors *in vivo* and *in vitro* [37]. 

Quorum sensing (QS) system controls the expression of aerolysin and numerous other virulence factors and biofilm development [38,39]. Blocking the action of aerolysin and biofilm by inhibiting QS has been shown in prior research to reduce the pathogenicity of *A. hydrophila* [40]. Additionally, developing novel and quick molecular docking techniques has enhanced molecular simulations with critical applications for screening and drug discovery [41,42,43]. A useful approach in drug design and compound screening for the development of natural drugs is the study of molecular docking of protein–ligand interactions [44,45]. It is possible to anticipate the conformations and binding affinities of the putative phytoconstituents from the extracts. This research will focus on the successful development of new medications by screening agricultural waste-derived metabolites for the diseases caused by *A. hydrophila* in aquaculture industries.

## 2. Results

### 2.1. Scanning Electron Microscopy

SEM images demonstrated a decrease in the biofilm development of *A. hydrophila* when treated with agricultural waste extracts. A maximum cell size and shape reduction was seen at the treatment concerning the calculated minimal inhibitory concentration (MIC) when control images revealed a cell matrix. Streptomycin (50 µg/mL) was used as a positive control and it exhibited dispersed cells without biofilm formation and morphological changes of the *A. hydrophila* cells (Figure 1 and Figure 2). 

### 2.2. FT-IR Analysis of Bacterial Biomass

The FT-IR spectra of bacterial biomass treated with black gram pod extracts and groundnut shell extracts (Figure 3 and Figure 4). The peak at 3350–3450 cm^−1^ indicates the existence of the OH group, which contains carbohydrates, proteins, and polyphenols and is classified as an alcoholic group, and a minor intensity peak at 1700–1715 cm^−1^ indicates the presence of fatty acid groups. Certain peaks at 3000 cm^−1^ show the presence of C-H alkenes and aromatic rings as a result of the interaction of the metabolites present in the ethyl acetate, acetone, methanol, and ethanol extracts of both groundnut shells and black gram pods. In contrast, non-polar solvents such as petroleum ether and hexane did not show any clear bands.

### 2.3. Homology Modelling of AhEUS112 Aerolysin

Homology (comparative) modelling is typically considered the most reliable *in silico* technique for predicting accurate 3D protein models using amino acid sequences [46,47]. The best AhEUS112 aerolysin sequence model (Figure 5a) with the lowest DOPE (discrete optimized protein energy) score (Figure 5b) is chosen.

Using PROCHECK, the modelled structure is validated using a Ramachandran plot using the RAMPAGE server [48]. The Ramachandran plot of the modelled protein represents 89.1% (366 aa) of the total residues in the most favoured regions. In comparison, 10.7% (44 aa) are in further and generously allowed regions, and only 0.2% of residues are found in the disallowed region. Based on the Ramachandran plot, the modelled structure indicates a good quality model (Figure 6). As a result, the predicted structure is chosen for molecular docking and molecular dynamics simulations.

### 2.4. Phylogenetic Analysis of the Aerolysin

The AhEUS112 aerolysin shared 90–96% of its identity with other bacterial aerolysin when their multiple sequence alignment was analyzed and aerolysin from other *Aeromonas* sp. (Figure 7). MEGAX software was used to construct the distance matrix of the aerolysin sequence obtained from the different species [49]. 

### 2.5. Molecular Docking

The main goal of *in silico* docking analysis of this study was to identify the optimal binding conformations between aerolysin and metabolites from the agri-wastes that blocks the function and membrane potential. To interpret the optimal binding position for the ligands and the drugs developed, molecular docking was used to assess the great affinity for the aerolysin active site residues of the *A. hydrophila*. Based on this, several positions were created and evaluated. Crude extracts have both active and inactive chemical compounds in their mixture of diverse chemical molecules that exhibit high affinity and complementarity to the target protein. The capacity of the ligands to interact with the target protein *in vivo*, which impacts the outcomes of molecular docking studies, relates to the substances in crude extracts. However, the molecular docking study may only foresee a potent binding relationship, if the active components in the crude extract have low affinities or can efficiently access the target region in the protein [50,51]. The metabolites chosen from the different extracts of BGP and GNS for the interactions of aerolysin are shown in Table 1. 

The number of hydrogen bonds that interacted with the aerolysin and the residues involved in the interactions were given in Table 2. The H-Pyran-4-one-2,3 dihydro-3,5 dihydroxy-6-methyl showed the strongest affinity with aerolysin possessing binding energy of −5.3 (kcal/mol), followed by 2-Hexyldecanoic acid and 2,2-Difluorocycloheptan-1-one (−5.2 kcal/mol), Methyl alpha-D-glucopyranoside (−5.1 kcal/mol), 5-Hydroxymethylfurfural (−5.0 kcal/mol), Methyl-d-glucose and Palmitic acid (−4.9 kcal/mol), Ethyl linoleate (−4.6 kcal/mol), Pentanone-5-methoxy (−4.3 kcal/mol), Diacetone alcohol (−4.1 kcal/mol), Methyl palmitate (−3.9 kcal/mol), and Cyanoacetic acid (−3.6 kcal/mol), respectively. The number of hydrogen bonds found during the interactions of metabolites with the aerolysin was represented in (Figure 8 and Figure 9). 

### 2.6. Simulation Dynamics

Molecular dynamic simulation (MDS) was used to determine the precise interaction of the ligand candidates with the protein under investigation. A methodology involving molecular docking, molecular dynamics, and free energy computing was used to identify the properties of specific natural compounds in a solvation state. In the current study, a 100 ns MDS was used to determine the best-docked molecule of H-Pyran-4-one-2,3 dihydro-3,5 dihydroxy-6-methyl, and 2-Hexyldecanoic acid to the aerolysin based on binding affinity and conformational stability (Figure 10a–d).

A significant RMS fluctuation was found between aerolysin and the other two ligands up to 30 residues, then showed similar fluctuation with all three complexes throughout the protein residues. The root-mean-square deviation (RMSD) estimate of backbone atoms varied from 0.25 nm to a maximum of 1.5 nm across the whole simulation. The RMSD value of the protein aerolysin was increased to 1.5 nm (10 ns), then showed at 0.75 nm (22.5 ns), and maintained steadily at 1 nm (up to 100 ns). However, aerolysin interacted with H-Pyran-4-one-2,3 dihydro-3,5 dihydroxy-6-methyl exhibited at 1 nm (45 ns) and then maintained at 0.75 nm (until 100 ns). Similarly, aerolysin interacted with 2-Hexyldecanoic acid and possessed an RMSD value of 0.75 nm at 10 ns, and it was gradually increased and maintained at 0.75 nm (until 100 ns). This data showed that the aerolysin formed a stable complex with the H-Pyran-4-one-2,3 dihydro-3,5 dihydroxy-6-methyl and 2-Hexyldecanoic acid at the range of 0.75 nm steadily (from 25 ns to 100 ns). 

## 3. Discussion

The extraction of phenolic compounds depends on the nature of the solvents. Polar solvents have lower electrostatic interactions that easily interact with the compounds present in the plant extracts that interchange their functional groups [52]. However, non-polar solvents can easily penetrate bacterial cells due to their lower charge [53]. Gram-negative bacteria such as *A. hydrophila* have rigid cell membranes that prevent the entry of the compounds into the cytoplasm [54]. They also have lipopolysaccharides that limit the penetration of hydrophobic compounds [55]. Based on our previous results, both GNS and BGP solvent extracts possess phenols and tannins [56,57]. These primary bioactive compounds are the major cause that exhibits better antioxidant and antibacterial properties [58,59]. The metabolites from the polar solvents also tend to diffuse the fatty acid composition of the rigid layer of *A. hydrophila* [60]. The metabolites of plant extracts may act on reducing the colonization of body surfaces and different epithelial layers, certain inorganic and organic molecules, along with other micro and macronutrients which are necessary for cell growth also promotes cell adhesion [61]. After a 48 h treatment with the extracts of both GNS and BGP, *A. hydrophila* cells were shrunken. They underwent splitting due to metabolites such as palmitic acid, methyl linoleate, H-Pyran-4-one-2,3 dihydro-3,5 dihydroxy-6-methyl, and 2-Hexyldecanoic acid [62,63,64,65]. These metabolites adhered to the lipopolysaccharides of the cell membrane, thus altering the bacterial cell morphology [66]. The metabolites from the extracts may inhibit nutrient availability that paved the way for bacterial cell growth [67]. The formation of the matrix by bacterial cells was separated due to inhibiting quorum-sensing signals from one cell to another [39,68]. This QS controls the synthesis of exopolysaccharide (EPS) by the *A. hydrophila* [69,70]. These polysaccharides, proteins, and nucleic acids played a crucial role in preventing the entry of antimicrobial agents and antibiotic exposure [71,72]. These exopolysaccharides play a vital role in cell detachment, colonization, and safeguarding forces of bacterial cells. The approaches to developing the new drug to combat multi-drug resistance and tolerance by polysaccharide lyases, a key enzyme which targeting the production of exopolysaccharides. Reduction in the exopolysaccharide production affects the QS signals between the cells [73,74]. According to Pellock et al. [75], *expr* is the major gene that maintains the quorum-sensing mechanism, and it is a homologue to *lux* receptors that leads to controls the production of exopolysaccharides. However, gram-negative bacteria such as *A. hydrophila* had autoinducers that tend to diffuse in and out of the cell [76]. These autoinducers, such as acyl-homoserine lactones (AHLs) synthesized by S-adenosylmethionine, bind to the cytoplasmic receptors and regulate the quorum-sensing gene expression [77,78,79]. Interfering with the synthesis, transport, or identification of autoinducers can be used to prevent quorum sensing. The key strategy is to utilize quorum-sensing inhibitors, which imitate or interfere with autoinducer binding to their receptors. These metabolites can impair quorum sensing in several bacterial species and limit biofilm development [80]. In gram-negative bacteria such as *A. hydrophila*, LuxR-type cytoplasmic receptors interact with another cell by detecting the AHLs; this complex transfers the quorum signals [81]. Additionally, fatty acids inhibit energy generation and cell lysis by interfering with components and preventing food intake [82]. Several studies have investigated the effects of fatty acids on mixed culture biofilms in the presence of natural conditions that may affect microbial signal production and reception [83]. 

In the FT-IR spectrum, the intensity peak at 1120–1160 cm^−1^ indicates the presence of polysaccharides in both control and treated groups due to bacterial biofilm formation [84,85]. Peaks obtained in the 2800–2600 cm^−1^ confirm the presence of aldehydes in the extract-treated biomass compared to the control [86]. The fatty acid groups in the polar and mid-polar extracts interact with the electron transport chain of bacteria. It involves the ATP transfer, which inhibits the bacterial enoyl-acyl reductase and leads to bacterial death [87]. In microbes, the electron acceptor is oxygen; when it demands, the organism tends to find an alternative to accept in the form of oxidized metals or non-metals [88]. During oxygen depletion, *A. hydrophila* utilizes iron (III) as an electron acceptor [89,90]. Carbon dioxide formed during the reactions will generate electrons which are accepted by iron (III) [91,92]. In our study, it is suggested that the metabolites present in the extracts of the GNS and BGP inhibit electron transfer by directly inhibiting iron reductase in the complex reactions. The antibacterial nature of the metabolites is based on solvent extraction [93]. Non-polar solvents are chemically inert and do not mix with water, so the microorganism can easily grow in the watery phase [94]. Essentially, polyphenols and bacteria interact in a non-specific manner, relying on the hydrogen group and hydrophobic effects that may have a significant influence owing to lipophilic interactions and the creation of covalent bonds [95]. Phenolic compounds present in the extracts may directly interact with the bacterial cell membrane, which causes intracellular leakage and ROS generation [96]. FT-IR analysis of *A. hydrophila* biomass can reveal important information about the bacterium’s chemical composition, such as the presence of proteins, lipids, and fatty acids [97]. This knowledge can help us understand the structure and function of the bacterial cell, as well as create ways to prevent or treat *A. hydrophila* infections. An ideal tree was generated by utilizing the neighbour-joining method to analyze the evolutionary history of the aerolysin of *A. hydrophila* [98]. The existence of several branches representing the different architectural structures of a protein was evident in the phylogenetic tree created from the multiple sequence alignment of the AhEUS112 aerolysin with aerolysin from 200 different bacterial species [99]. It is a feasible approach to find protein areas that have been conserved during evolution by comparing the sequences of various species [100]. These conserved regions of the protein may be critical for protein function and might be targeted for drug development or other purposes.

A mixture of hydrophobic and van der Waals interactions with active site residues also stabilized the ligand configurations [101,102]. To emphasize, amino acid characteristics impact the functional activities of certain residues based on the physicochemical restrictions to variation of amino acid position/alignment [103]. The data analysis showed that aerolysin had common interaction residues with most test compounds.

Molecular docking and homology modelling were unique and useful tools for characterizing protein–ligand interaction patterns in configuration [104]. Due to the strong covalent bonds, weak intermolecular linkages encompassed a variety of interactions that did not involve the exchange of electrons. Still, hydrogen bonds played a vital role in the interaction of proteins and ligands [102]. GRID detects favourable sites for ligand binding with protein [105]. The binding nature between the ligand and protein depends on the length and orientation [101,106]. These protein–ligand interactions formed due to the cavity shape, size, and energy level of pocket formation [107,108]. Ligands are compounds that can control the activity of a protein or enzyme by binding to specific sites on the target protein or enzyme. In the case of aerolysin, ligands can be employed to prevent the production of toxin aggregates, which can injure host cells and tissues [109]. Ligands can bind to different sites of aerolysin, such as hydrophobic regions on the toxin’s surface, particular spots on the pore-forming domain, and other sections of the molecule [110]. Flavonoids and polyphenols have been demonstrated to suppress the production of aerolysin aggregates. These bioactive compounds can attach to particular sites on the toxin and prevent it from building huge complexes that can damage host cells. This kind of *in vitro* approach is beneficial in decreasing aerolysin toxicity [111]. Our docking studies showed that the key residues of the aerolysin protein’s binding pocket, such as Tyr 337, Arg 417, Arg 414, and Tyr 135 interacted with pentanone-5-methoxy via traditional hydrogen bonding and hydrophobic interactions [112]. These findings stated that pentanone-5-methoxy might reduce quorum sensing by decreasing the expression of aerolysin, which then affects other virulence-associated genes.

Palmitic acid reacted significantly with aerolysin, with a binding energy of −4.3 kcal/mol and a two-hydrogen bonding interaction (Arg 379, Ile 378, Ser 377, Phe 371, Tyr 380); these results agree with studies that reported the palmitic acid inhibiting the virulence factors associated with biofilm [113,114]. According to Dong et al. [115], heptamer formation was controlled by the ARG 414 and ARG 417 residual movements. This was the basic action mechanism behind the inhibition of aerolysin by the ligand H-Pyran-4-one-2,3 dihydro-3,5 dihydroxy-6-methyl. However, 2-Hexyldecanoic acid is bound with the ASP360 and does not involve forming heptamer [19]. Aerolysin often had the propensity to form a heptamer after entering the host cell membrane [116]. This heptamer had a transmembrane pore that affected the permeability of the host cell membrane and caused cell death [117]. The flexible portion of a protein or the parts of structures that change concerning the overall structure was evaluated by the root-mean-square fluctuation (RMSF) [118]. 

The simulation’s dynamics give scientists a unique perspective on the structural and functional changes that occur during ligand binding by allowing them to watch the movement of specific atoms in the protein and the ligand over time [119]. The radius of gyration of aerolysin, protein complex with 2-Hexyldecanoic acid, and protein complex with the H-Pyran-4-one-2,3 dihydro-3,5 dihydroxy-6-methyl was determined. Using thermodynamic concepts, the radius of gyration indicated the protein’s compactness with protein folding and unfolding [120]. The radius of gyration cannot be precisely measured because of diverse samples [121]. The Rg values were obtained in the range of 2.75–3.0 nm, whereas the aerolysin was maintained at 3.2 nm, and the aerolysin complexed with 2-Hexyldecanoic acid lay at 3.2 nm, respectively, which gradually increased and maintained at 3.4 nm from 40–100 ns. However, the aerolysin with H-Pyran-4-one-2,3 dihydro-3,5 dihydroxy-6-methyl showed an Rg value at 3.4 nm initially and it held at 3.6 nm. With this evidence, the examination of dynamics’ mean radius of gyration fell within the range of random-coil statistics, confirming the protein folding in the presence of residual structure [122]. According to studies, aerolysin pores are fairly far from the host membrane surface and are shown as nanodisc-entrapped pores compatible with the absence of hydrophobicity [123]. The molecular dynamics trajectories for the whole examined protein–ligand complex is typically stable and within acceptable limits for the 100 ns simulation period, according to the RMSD fluctuation analysis [124]. According to the findings, the inhibitor attaches to a particular site of the protein and stabilizes it in a closed conformation, preventing the formation of the opening in the membrane [125]. Overall, the results of the molecular dynamics simulation study imply the stability of the protein–ligand complex of aerolysin with metabolites from agricultural waste.

## 4. Materials and Methods

### 4.1. Maintenance of Bacterial Strain and Culture Media Preparation

A fish pathogen, *A. hydrophila* (glycerol stock preserved at Laboratory of Aquabiotics/Nanoscience, Bharathidasan University, Tiruchirappalli, Tamil Nadu, India) (35 ± 2 °C/24 h), and bacterial cultures were maintained and grown in Tryptic soy agar or broth (TSA/TSB) containing Tryptone 1%, yeast extract 0.5%, and sodium chloride 0.5% with 1.2% agar.

### 4.2. Preparation of Extract

Groundnut shells (GNS) and black gram pods (BGP) were collected based on their detailed experimental procedures [56,57]. The collected agri-wastes were shade dried at 37 °C, ground into a coarse powder, sieved using 0.2 mm sieve plates, and then stored at −20 °C for subsequent examination in an airtight container. Cold maceration was used to elution the extracts from powdered agri-wastes using six solvents (10:90 *w*/*v*): ethyl acetate, petroleum ether, methanol, ethanol, hexane, and acetone. Additionally, the solvents employed for this study is based on the polarity which ranges from least polar to most polar [126]. The extracts were concentrated at roughly 40 °C in a rotating vacuum evaporator under decreasing pressure until agglomerates were formed; before that, the filtrate was collected using Whatman No. 1 filter paper. The extracts were dried to remove excess solvents and kept at 4 °C for future research. For experimental purposes, dry extracts were reconstituted with DMSO (0.1%) [127].

### 4.3. Scanning Electron Microscopy

The inhibition and deterioration of *A. hydrophila* biofilm using the various extracts of GNS and BGP were visualized using scanning electron microscopy (SEM) with a slight modification of the detailed protocol by Zhou et al. [128]. In short, biofilms of *A. hydrophila* grown on glass coverslips (18 mm) submerged in nutrient broth with determined minimum inhibitory concentration (MIC) were given in Table 3. Based on this, various extracts (acetone, methanol, hexane, ethanol, ethyl acetate, and petroleum ether) of groundnut shells and black gram pods were poured into six-well plates, and the untreated (without extracts) acted as the negative control. Streptomycin (50 µg/mL) was used as a positive control. The treated and untreated plates were incubated for 48 h at 37 °C, and then gently washed with miliQ to extract adherent bacterial cells. Samples were kept in 2.5% glutaraldehyde for 15 min and dehydrated with 25–95% gradient ethanol for 10 min. The dried biofilms were gold coated and examined under a scanning electron microscope (SEM, TESCAN, Czech Republic, and Vega 3).

### 4.4. FT-IR Analysis of Bacterial Biomass

The bacterial cells of *A. hydrophila* treated with the extracts of agricultural wastes were collected through centrifugation at 10,000 rpm for 10 min and washed with phosphate buffer, pH 7.0, then made into a die using a desiccator at 45 °C. The KBr crystals were vacuum-dried as described [129]; 1000 mg of KBr and 2.5 mg of bacterial biomass were finely powdered and homogenized. The KBr beta press was used to form 100 mg of this bulk mixture into a single pellet. With the bacterial biomass abundant as a clear pellet within the KBr beta press barrel, the barrel was put on the sample holder in the FT-IR chamber. Fourier transform infrared (FT-IR) spectrophotometer (Perkin Elmer, Waltham, MA, USA) (4000–500 cm^−1^) scans were then performed, and the FT-IR chamber was carefully modified until water vapour peaks were eliminated [130].

### 4.5. Ligand Screening for Molecular Docking

This ligand screening is based on the metabolites identified through GC-MS analysis from various solvent extracts of groundnut shells and black gram pods from our previous study [56,57]. A total of 325 metabolites were identified, of which 14 compounds from groundnut shells and 20 compounds from black gram pods with potential antibacterial efficacy were chosen for this study based on the earlier literature to analyze their interaction with the aerolysin (Table 1). The chemical structure of each drug/compound was retrieved in structure-data file (SDF) format from the PubChem database (https://pubchem.ncbi.nlm.nih.gov/; accesed on 9 March 2021), and Open Babel was used to convert SDF to mol2 format [131].

### 4.6. Phylogenetic Analysis of Aerolysin

A multiple sequence alignment with the AhEUS112 aerolysin amino acid sequence was performed on the amino acid sequences obtained from 200 various bacterial species (*Aeromonas* sp. and other related bacteria). The evolutionary analysis was constructed with a neighbour-joining (NJ) algorithm using the MEGAX maximum likelihood method [98,132]. In the bootstrap test, the numbers next to the branches indicated the fraction of duplicate trees in which the related taxa were clustered together (100 repetitions) [133]. The tree was built using the maximum likelihood method and visualized using iTOL (Interactive Tree of Life) (https://itol.embl.de/; accessed on 24 April 2023). The phylogenetic tree’s branch lengths were shown to scale and correspond to the evolutionary distances. The number of amino acid changes per site was used to calculate evolutionary distances using the P-distance approach [134]. All unclear places for each sequence pair were eliminated, leaving a final data set of 523 positions that were utilized for analysis.

### 4.7. Structural Analysis of Aerolysin

The AhEUS112 aerolysin’s amino acid sequence was analyzed using BLAST-P to find the most appropriate template for homology modelling (accession no. MT491733) [135]. Following a similarity search for the best-aligned aerolysin crystal structures published in the Protein Data Bank (PDB), 1HWG (PDB ID) was selected for the modelling template. MODELLER software was used to construct the 3D model of the target sequence and structure validated with the Ramachandran plot using the SAVES server.

### 4.8. Molecular Docking and Simulation Dynamics

The potential ligands identified from GNS and BGP with aerolysin were used for molecular docking through AutoDock software [136,137]. The compounds with the highest binding affinity were chosen for the best-docked complexes. Using Webgro (online server), the modelled protein aerolysin was subjected to a molecular dynamics simulation. Using OPLS forcefield, MD simulation of both ligands and protein was analyzed for 100 ns [138,139]. 

## 5. Conclusions

This is a kind, sensible, and effective tactic in anti-virulence treatment, which involves employing different aerolysin inhibitors or substances that lead to preventing QS in bacterial pathogens such as *A. hydrophila*. Extracts from agricultural waste, such as groundnut shells and black gram pods, have been evident in SEM micrographs to block QS signals and disrupt the growth of biofilms. Additionally, the 3D structure of aerolysin has been generated, and it plays a major role in causing septicemia. Using an *in silico* technique in this study, H-Pyran-4-one-2,3 dihydro-3,5 dihydroxy-6-methyl and 2-Hexyldecanoic acid are shown to be more effective in inhibiting aerolysin oligomerization of *A. hydrophila.* This protein homology implies that a different potential pharmacological target could possibly work to restrict the activity of aerolysin in other pathogenic bacteria to form biofilms. It also provides novel insight that limits the broad usage of pharmaceutical drugs for *in vitro* testing. Thus, the agricultural waste extracts could be used as an appropriate medicine to block aerolysin activity by *A. hydrophila*, and they may aid in treating hemorrhagic septicemia. The outcome of this study enlightens the aquafarmer and the country’s economy by overcoming the major disease outbreak in aquaculture by *A. hydrophila*.

## Figures and Tables

**Figure 1 antibiotics-12-00891-f001:**
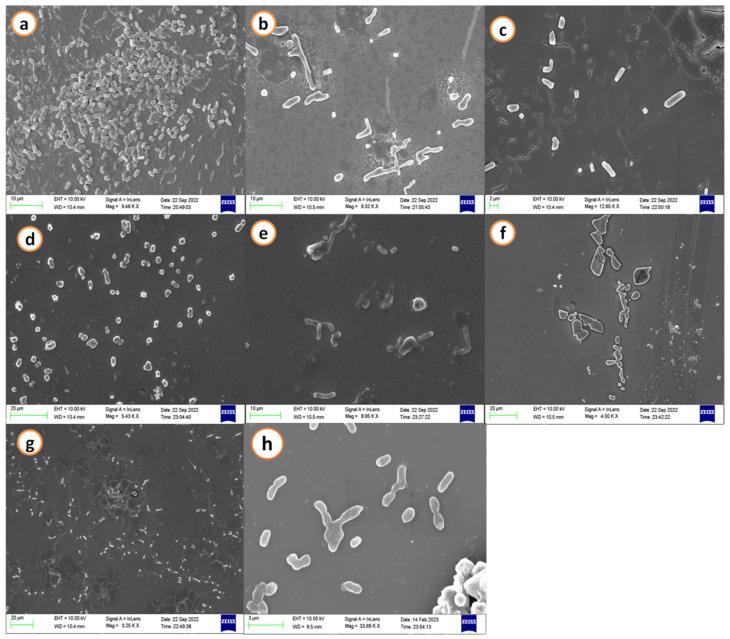
Scanning electron microscopic images of *A. hydrophila* biofilm matrix. (**a**) Negative control: shows dense and thick adherence of biofilm; treated with groundnut shell extracts (**b**) methanol, (**c**) ethanol, (**d**) acetone, (**e**) ethyl acetate, (**f**) hexane, (**g**) petroleum ether, and (**h**) positive control: streptomycin (50 µg/mL) exhibits dispersed biofilm.

**Figure 2 antibiotics-12-00891-f002:**
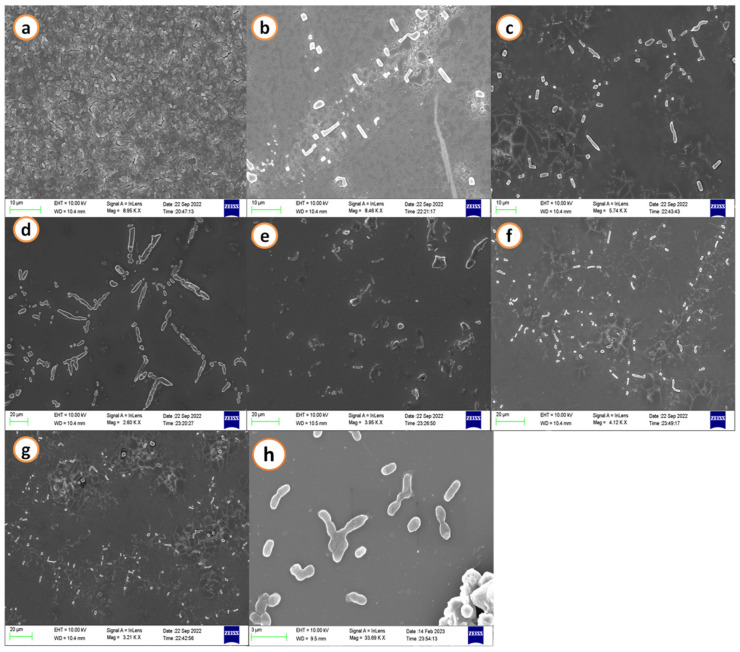
Scanning electron microscopic images of *A. hydrophila* biofilm matrix. (**a**) Negative control: shows dense and thick adherence of biofilm; treated with black gram pod extracts (**b**) methanol, (**c**) ethanol, (**d**) acetone, (**e**) ethyl acetate, (**f**) hexane (**g**) petroleum ether, and (**h**) positive control: streptomycin (50 µg/mL) exhibits dispersed biofilm.

**Figure 3 antibiotics-12-00891-f003:**
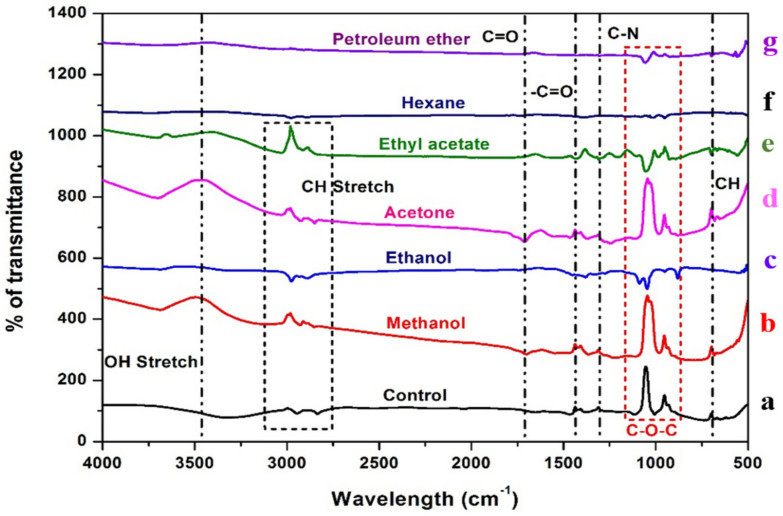
FT-IR spectra of the *Aeromonas hydrophila* biomass treated against various extracts of groundnut shell. Red dotted lines indicate the presence of polysaccharides. Untreated: (**a**) control. Treated: (**b**) methanol, (**c**) ethanol, (**d**) acetone, (**e**) ethyl acetate, (**f**) hexane, and (**g**) petroleum ether.

**Figure 4 antibiotics-12-00891-f004:**
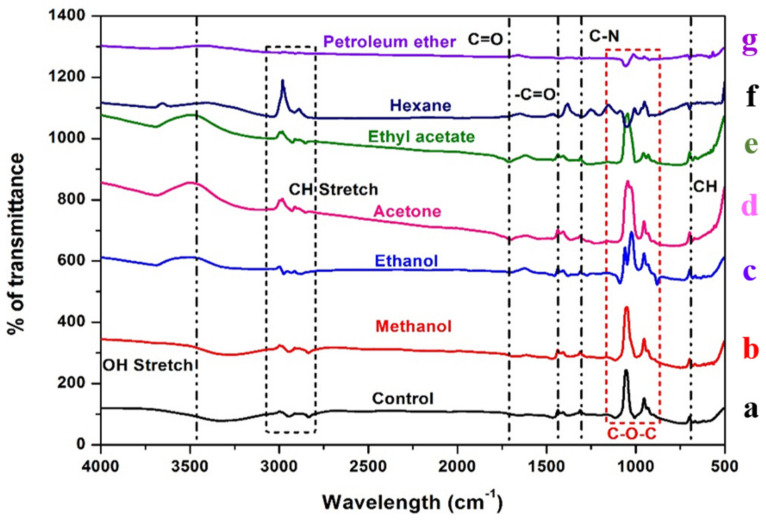
FT-IR spectra of the *Aeromonas hydrophila* biomass treated against various extracts of black gram pods. Red dotted lines indicate the presence of polysaccharides. Untreated: (**a**) control. Treated: (**b**) methanol, (**c**) ethanol, (**d**) acetone, (**e**) ethyl acetate, (**f**) hexane, and (**g**) petroleum ether.

**Figure 5 antibiotics-12-00891-f005:**
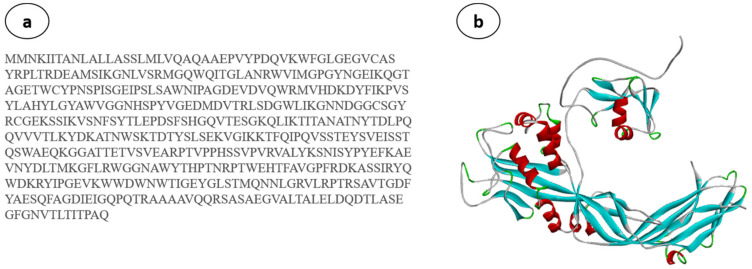
(**a**) Sequence of the AhEUS112 aerolysin. (**b**) Predicted three-dimensional structure of aerolysin from *A. hydrophila*.

**Figure 6 antibiotics-12-00891-f006:**
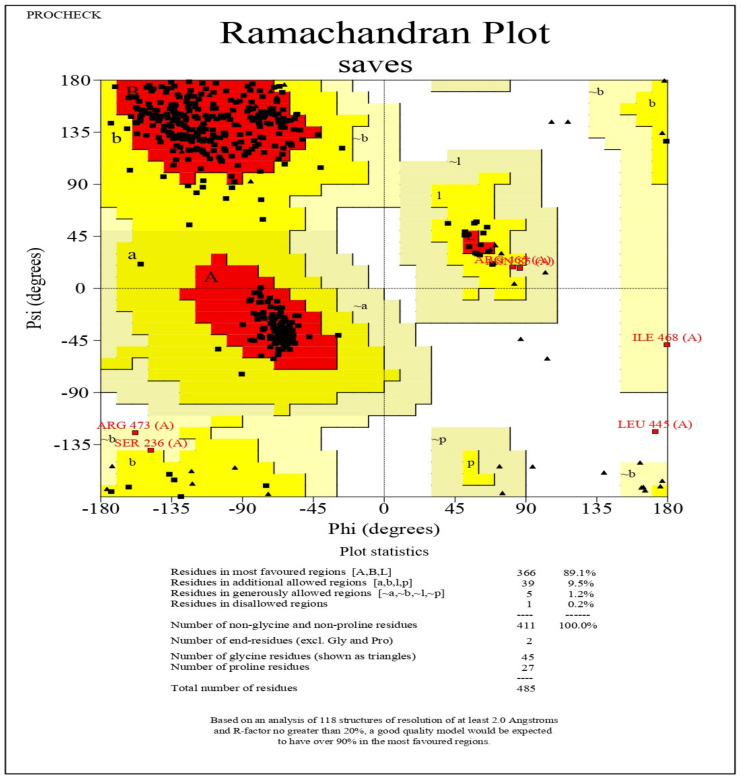
The stereochemical spatial arrangement of amino acid residues in the preferred area of the Ramachandran plot of the modelled 3D structure of aerolysin (Red coloured squares indicate residues in most favoured regions, dark yellow-coloured squares indicate the residues in additional allowed regions and pale yellow coloured square indicates residues in the generously allowed regions).

**Figure 7 antibiotics-12-00891-f007:**
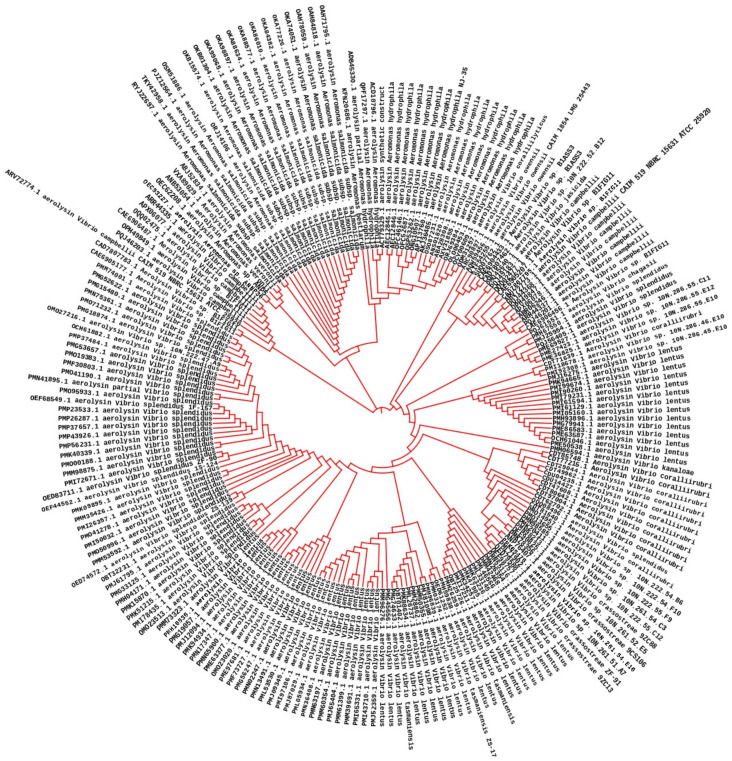
Phylogenetic tree of aerolysin from various species of *Aeromonas hydrophila*.

**Figure 8 antibiotics-12-00891-f008:**
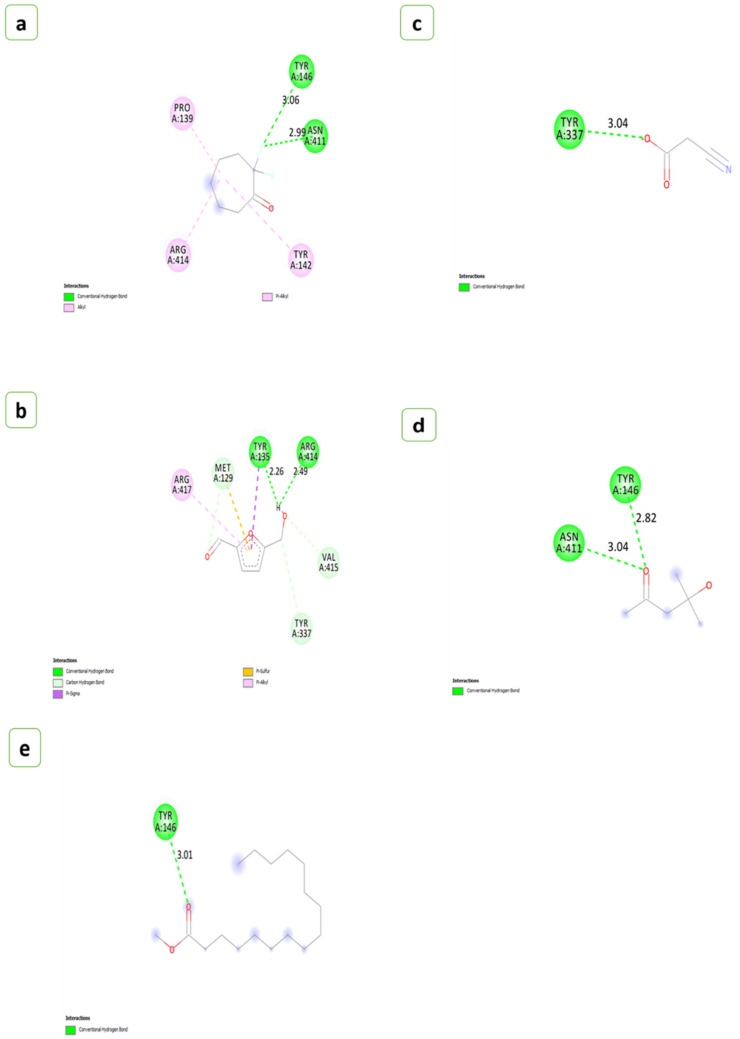
A 2D view of the interactions of the compounds extracted from black gram pods. (**a**) 2,2-Difluorocycloheptan-1-one, (**b**) 5-Hydroxymethylfurfural, (**c**) Cyanoacetic acid, (**d**) Diacetone alcohol, (**e**) Methyl palmitate.

**Figure 9 antibiotics-12-00891-f009:**
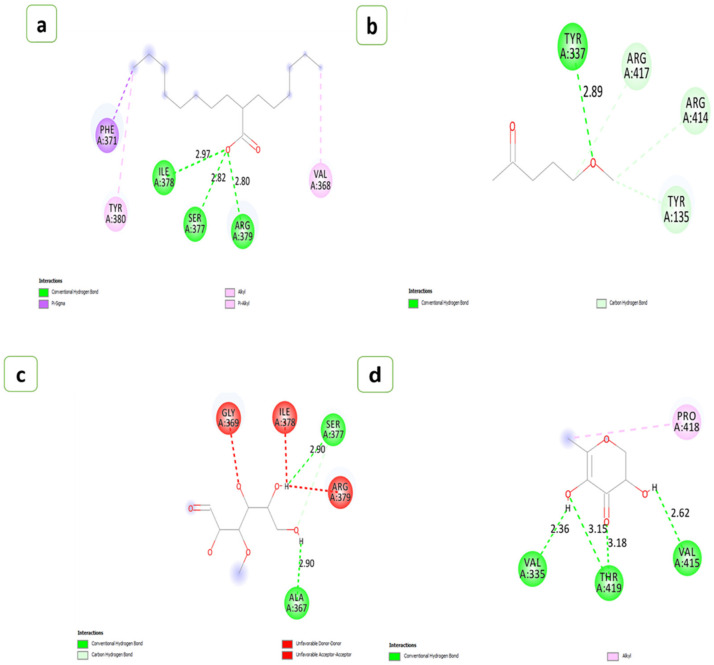
A 2D view of the interactions of the compounds from groundnut shells. (**a**) 2-Hexyldecanoic acid, (**b**) Pentanone-5-methoxy, (**c**) Methyl-d-glucose, (**d**) H-Pyran-4-one-2,3 dihydro-3,5 dihydroxy-6-methyl, (**e**) Ethyl linoleate, (**f**) Methyl alpha-D-glucopyranoside, (**g**) Palmitic acid.

**Figure 10 antibiotics-12-00891-f010:**
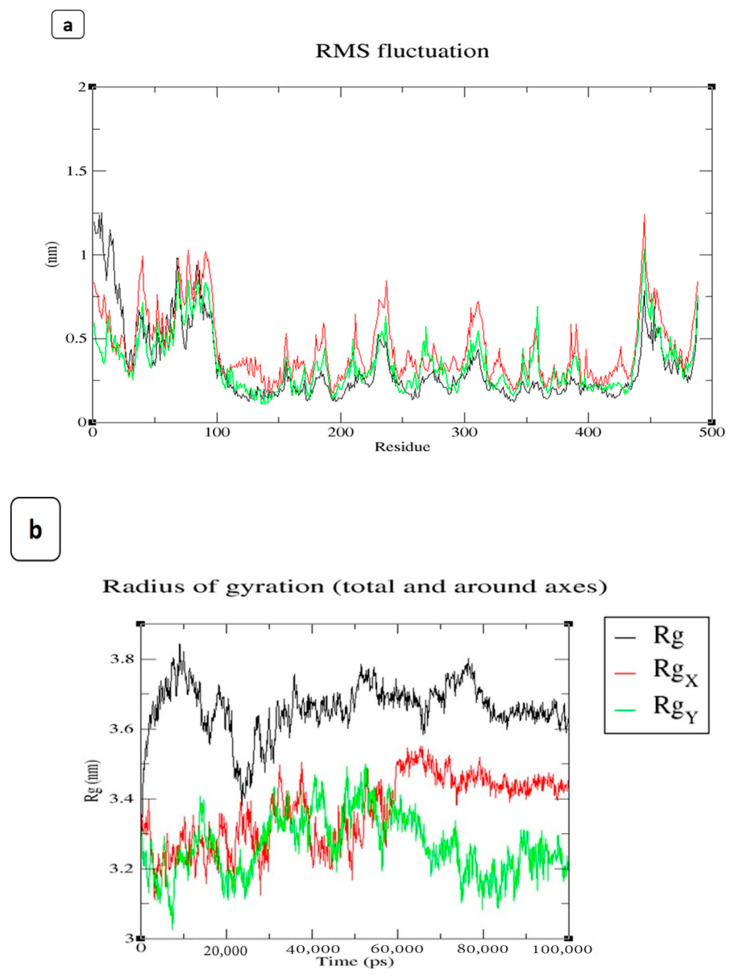
Graphical representation of 100 ns MD simulation analysis of aerolysin (in black colour), a protein with H-Pyran-4-one-2,3 dihydro-3,5 dihydroxy-6-methyl (in green colour), and protein with 2-Hexyldecanoic acid (in red colour). (**a**) RMSF values of the backbone atoms, (**b**) radius of gyration of the backbone atoms, (**c**) RMSD values of Cα atoms in the protein, and (**d**) hydrogen bonds stability of the protein and complexes.

**Table 1 antibiotics-12-00891-t001:** List of compounds for molecular docking with aerolysin from black gram pods and groundnut shells.

Black Gram Pods		Groundnut Shells	
S. No.	Name of the Compound	S. No.	Name of the Compound
1.	1-Hexadecene	1.	2-Hexyldecanoic acid
2.	1-Isopropoxy-2-propanol	2.	2-Pentanone, 5-methoxy
3.	2,2-Difluorocycloheptan-1-one	3.	3-O-Methyl-d-glucose
4.	3-7-11-15-Tetramethyl-2-hexadecen-1-O	4.	4H-Pyran-4-one- 2-3-dihydro-3-5-dihydroxy-6-methyl
5.	5-Hydroxymethylfurfural	5.	Cyclohexanone
6.	Azulene	6.	Eicosane
7.	Butyronitrile	7.	Ethyl linoleate
8.	Cholesterol propionate	8.	Hexatriacontane
9.	Cholesterol	9.	Methyl alpha-D-glucopyranoside
10.	Cyanoacetic acid	10.	Octadecane
11.	Diacetone alcohol	11.	Palmitic acid
12.	Dodecanel	12.	Pentadecane- 2-6-10-13-tetramethyl
13.	Heptadecane	13.	Stearic acid
14.	Hexadecane	14.	Tetracosane
15.	Methyl palmitate		
16.	Methyl propyl ether		
17.	Naphthalene		
18.	Tetracontane		
19.	Tetratetracontane		
20.	Z-5-Nonadecene		

**Table 2 antibiotics-12-00891-t002:** Molecular docking of aerolysin with metabolites identified from both black gram pod and groundnut shell extracts. (Arg-Arginine, Pro-Proline, Tyr-Tyrosine, Asn-Asparagine, Met-Methionine, Val-Valine, Ile-Isoleucine, Ser-Serine, Phe-Phenylalanine, Gly-Glycine, Lys-Lysine).

S. No.	Compound	Binding Energy (kcal/mol)	Hydrogen Bond Interactions	Residues Involved During Interactions
1	2,2-Difluorocycloheptan-1-one	−5.2	2	Arg 414, Pro 139, Tyr 146, Asn 411, Tyr 142
2	5-Hydroxymethylfurfural	−5.0	5	Arg 417, Met 129, Tyr 135, Arg 414, Val 415, Tyr 337
3	Cyanoacetic acid	−3.6	1	Tyr 337
4	Diacetone alcohol	−4.1	2	Asn 411, Tyr 146
5	Methyl palmitate	−3.9	1	Tyr 146
6	2-Hexyldecanoic acid	−5.2	3	Phe 371, Tyr 380, Ile 378, Ser 377, Arg 379, Val 368
7	Pentanone-5-methoxy	−4.3	4	Tyr 337, Arg 417, Arg 414, Tyr 135
8	Methyl-d-glucose	−4.9	2	Gly 369, Ile 378, Ser 377, Arg 379, Ala 369
9	H-Pyran-4-one-2,3 dihydro-3,5 dihydroxy-6-methyl	−5.3	4	Val 335, Thr 419, Val 415, Pro 418
10	Ethyl linoleate	−4.6	1	Arg 414, Pro 139, Tyr 142, Lys 138
11	Methyl alpha-D-glucopyranoside	−5.1	4	Leu 416, Thr 419, Glu 334
12	Palmitic acid	−4.9	3	Arg 379, Ile 378, Ser 377, Phe 371, Tyr 380

**Table 3 antibiotics-12-00891-t003:** Determination of minimum inhibitory concentration (MIC) of groundnut shell and black gram pods for *A. hydrophila* (µg/mL).

S. No..	Extraction Solvents	Minimum Inhibitory Concentration (MIC) for *Aeromonas hydrophila* (µg/mL)	
		Groundnut Shell	Black Gram Pod
1.	Methanol	250	250
2.	Ethanol	250	250
3.	Acetone	500	500
4.	Ethyl acetate	500	500
5.	Hexane	500	500
6.	Petroleum ether	500	500

## Data Availability

Data will be made available on reasonable request.

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
