# Peer review of "Evaluating Biofilm Inhibitory Potential in Fish Pathogen, Aeromonas hydrophila by Agricultural Waste Extracts and Assessment of Aerolysin Inhibitors Using In Silico Approach"

_antibiotics, 2023, doi:10.3390/antibiotics12050891_

Round 1

Reviewer 1 Report

The authors presented a manuscript to investigate the effect of crude solvent extracts of both groundnut shells and black gram pods on biofilm inhibitory of A. hydrophila. And they used molecular docking to analyst the potential hydrogen bond interactions with AerA. My comments are in following:

Major comments:

1)       The introduction should to be divided into paragraphs according to the content.

2)       I suggest the author add experiments to investigate the antibacterial effect of solvent extracts

3)       The solvent exact composition of the extract needs to be analyzed.

4)       What is the relationship between the results of Molecular Docking analysis and the components in the crude extracts.

Minor comments:

1)       kcal/mol-1 to kcal/mol (line33, Table 2 )

2)       Please keep the names of bacteria consistent.

3)       The legend of Figure 10 should be placed at the end of figure.

4)       Delete Line334-337

Author Response

Comments and Suggestions for Authors

Reviewer 1:

Major comments: 

  • The introduction should to be divided into paragraphs according to the content.

Author’s response: Based on the reviewer’s comments, the introduction section was divided into paragraphs as per the contents in the revised manuscript.

  • I suggest the author add experiments to investigate the antibacterial effect of solvent extracts

Author’s response: We thank the reviewer’s valuable suggestion. The antibacterial effects of both the groundnut shell and black gram pod solvent extracts were investigated and published already in the below mentioned research articles. The experiments such as agar well diffusion method, minimum inhibitory concentration determination, and bacterial growth curve kinetics were used to analyze the effects of solvent extracts against Aeromonas hydrophila. In this present manuscript, we focused to evaluate the extracts’ biofilm inhibitory potential and used in silico approach to identify inhibitors against virulence factor of Aeromonas hydrophila aerolysin. Our previous publications were also cited in the revised manuscript to provide the preliminary activity of solvents extracts against Aeromonas hydrophila.

The publications are:

  1. Arumugam, M., Manikandan, D.B., Sridhar, A., Palaniyappan, S., Jayaraman, S. and Ramasamy, T., 2022. GC–MS Based Metabolomics Strategy for Cost-Effective Valorization of Agricultural Waste: Groundnut Shell Extracts and Their Biological Inhibitory Potential. Waste and Biomass Valorization, 13(10), pp.4179-4209. https://doi.org/10.1007/s12649-022-01768-z

  1. Arumugam, M., Manikandan, D.B., Mohan, S., Sridhar, A., Veeran, S., Jayaraman, S. and Ramasamy, T., 2022. Comprehensive metabolite profiling and therapeutic potential of black gram (Vigna mungo) pods: conversion of biowaste to wealth approach. Biomass Conversion and Biorefinery, pp.1-32. https://doi.org/10.1007/s13399-022-02806-5

  • The solvent exact composition of the extract needs to be analyzed.

Author’s response: Author’s response: We thank the reviewer’s valuable suggestion, the major bioactive compounds present in the solvent extracts of both GNS and BGP were analyzed and published (above mentioned). The composition of the extracts was discussed with citations of our previous publications in page number 16; line number 240-254 & page number 17; line number 291-299 in the revised manuscript. The cited references have also been listed in the bibliography section (Reference No: 54-78).

Based on our obtained previous results, both GNS and BGP solvent extracts possess both phenols and tannins [54-55]. These primary bioactive compounds are the major cause which exhibit better antioxidant and antibacterial properties [56-57]. Along with that the metabolites from the polar solvents have the tendency to diffuse the fatty acid composition of the rigid layer of A. hydrophila. The metabolites from the polar solvents also tend to diffuse the fatty acid composition of the rigid layer of A. hydrophila [58]. The metabolites of plant extracts may act on reducing the colonization of body surfaces and different epithelial layers, certain inorganic and organic molecules. Other micro and macronutrients which are necessary for cell growth, also promote cell adhesion [59]. After 48 h treatment with the extracts of both GNS and BGP, A. hydrophila cells were shrunken and underwent splitting due to the presence of metabolites like palmitic acid, methyl linoleate, H-Pyran-4-one-2,3 dihydro-3,5 dihydroxy-6-methyl and 2-Hexyldecanoic acid [60-63]. These metabolites adhered to the lipopolysaccharides of the cell membrane, thus altering the bacterial cell morphology [64]. The metabolites from the extracts may inhibit nutrient availability that paved the way for bacterial cell growth [65]. The formation of the matrix by bacterial cells was separated due to the inhibition of quorum sensing signals from one cell to another [39, 66]

Essentially, Polyphenols and bacteria interact in a non-specific manner, relying on the hydrogen group and hydrophobic effects that may have a significant influence owing to lipophilic interactions and the creation of covalent bonds [93]. Phenolic compounds present in the extracts may directly interact with the bacterial cell membrane which causes intracellular leakage, and ROS generation [94].

4)       What is the relationship between the results of Molecular Docking analysis and the components in the crude extracts.

Author’s response: As per the reviewer’s comment, the exact relationship between the compounds found in the crude extracts and the molecular docking analysis were given in the revised manuscript. (Page number 8-9; line number 166-172). We had already identified the components of extracts and published. In this manuscript we made an attempt through molecular docking analysis to explore the rate of interactions of these compounds with the Aeromonas hydrophila virulence factor (aerolysin).

 It has both active and inactive chemical compounds in their mixture of diverse chemical molecules that exhibits high affinity and complementarity to the target protein. The capacity of the ligands to interact with the target protein in vivo which impact the outcomes of molecular docking studies relate to the substances in crude extracts. However, the molecular docking study may not foresee a potent binding relationship if the active components in the crude extract have low affinities or cannot efficiently access the target region in the protein [Reference no. 50-51]

Minor comments: 

  • kcal/mol-1 to kcal/mol (line33, Table 2 )

Author’s response: As per the reviewer’s comments, kcal/mol-1 has been modified to kcal/mol in the revised manuscript. The units were checked and changed accordingly in the entire revised manuscript to follow uniformity.

  • Please keep the names of bacteria consistent.

Author’s response: Based on the reviewer’s suggestion, the names of the bacteria have been given coherently in the revised manuscript.

3)       The legend of Figure 10 should be placed at the end of figure.

Author’s response: In view of reviewer’s comment, the legend for the figure 10 was placed at the end of the figure in the revised manuscript.

4)       Delete Line334-337

Author’s response: As per the reviewer’s comments, the line 334-337 have been deleted in the revised manuscript.

Reviewer 2 Report

In this Manuscript, Thirumurugan Ramasamy and co-authors studied the sensible and effective tactic in anti-virulence treatment, which involves employing different quorum-sensing inhibitors or substances that prevent QS in bacterial pathogens like Aeromonas hydrophilas. The authors were able to conclude that the agricultural- waste extracts could be used as an appropriate medicine to block aerolysin activity by A. hydrophila and also it may aid in the treatment of hemorrhagic septicemia. It is possible that the outcome of this study could enlighten the aquafarmer and the country’s economy by overcoming the major disease outbreak in aquaculture by Aeromonas hydrophila.

Besides this, the manuscript is well-written, the content is explained scientifically, and the data support the hypotheses. Based on the potential and strength of the present research, I would recommend it for publication in the Antibiotics without changes.

Author Response

Comments and Suggestions for Authors

Reviewer 2:

In this Manuscript, Thirumurugan Ramasamy and co-authors studied the sensible and effective tactic in anti-virulence treatment, which involves employing different quorum-sensing inhibitors or substances that prevent QS in bacterial pathogens like Aeromonas hydrophilas. The authors were able to conclude that the agricultural- waste extracts could be used as an appropriate medicine to block aerolysin activity by A. hydrophila and also it may aid in the treatment of hemorrhagic septicemia. It is possible that the outcome of this study could enlighten the aquafarmer and the country’s economy by overcoming the major disease outbreak in aquaculture by Aeromonas hydrophila.

Besides this, the manuscript is well-written, the content is explained scientifically, and the data support the hypotheses. Based on the potential and strength of the present research, I would recommend it for publication in the Antibiotics without changes.

Author’s response: We thank the reviewer for the valuable comments, and appreciation for our manuscript. Once again, we express the sincere thanks to the Editor and the Reviewer’s for their valuable comments and suggestions to improve the quality of the manuscript for possible publication.

Reviewer 3 Report

The manuscript is already in a good shape but the introduction and conclusion should remove some texts and present in a short and consize way.

There are also some minor mistakes. Such as Latin name should be Italic. there should be a pace between the number and the unit, such as 25ns should be 25 ns. The author should read the manuscript again carefully and fix the similar format issue. 

L137, should use the same font size

High resolution for Figure 7

L239, the Metabolites should be The

L241, quorum sensing should be QS

L261, the author should point out the IR spectrum 

delete L334-337

Author Response

Comments and Suggestions for Authors

Reviewer 3:

The manuscript is already in a good shape but the introduction and conclusion should remove some texts and present in a short and consize way.

There are also some minor mistakes. Such as Latin name should be Italic. there should be a pace between the number and the unit, such as 25ns should be 25 ns. The author should read the manuscript again carefully and fix the similar format issue. 

Author’s response: Based on the reviewer’s comments, the space between the unit and numbers were maintained uniformly throughout the revised manuscript. We checked the manuscript and the small errors were rectified. The other formats such as units, italics for scientific names were also corrected to maintain consistency in the revised manuscript.

L137, should use the same font size

Author’s response: As per the reviewer’s suggestion, the same font size was maintained throughout the revised manuscript.

High resolution for Figure 7

Author’s response: Based on the reviewer’s comments, the resolution of the figure 7 has been increased in the revised manuscript. Page number 8.

L239, the Metabolites should be The

Author’s response: In view of the reviewer’s comment, the typographical error was rectified and the sentence starts with “The” in the revised manuscript.

Page number 16; line number 252-253.

L241, quorum sensing should be QS

Author’s response: As per the reviewer’s comment, the quorum sensing has been abbreviated as “QS” in the revised manuscript.

Page number 16; line number 254.

L261, the author should point out the IR spectrum 

Author’s response: Based on the reviewer’s comment, the FT-IR spectrum have been included in the revised manuscript.

Page number 16; line number 277-279.

delete L334-337

Author’s response: As per the reviewer’s comment, the line 334-337 have been deleted in the revised manuscript.

Reviewer 4 Report

1-Please avoid acronyms and abbreviation in the abstract section : Groundnut shells (GNS) and black gram pods (BGP)..... 

2- Phylogenetic tree of Aerolysin from various species of A.hydrophil? this is your results. If is that the case the authors should include this work in material and method section 

Author Response

Comments and Suggestions for Authors

Reviewer 4:

1-Please avoid acronyms and abbreviation in the abstract section: Groundnut shells (GNS) and black gram pods (BGP)..... 

Author’s response: In view of the reviewer’s comments, the abbreviation for groundnut shells and black gram pods have been removed in the abstract of the revised manuscript.

Page number 1; line number 30.

2- Phylogenetic tree of Aerolysin from various species of A.hydrophila? this is your results. If is that the case the authors should include this work in material and method section

Author’s response: As per the reviewer’s comments, the software used for constructing the phylogenetic tree has already been given in the materials and methods

Section 4.6: Page number 20; line number 430-441.

Reviewer 5 Report

The work presented entitled “Evaluating biofilm inhibitory potential in fish pathogen, Aeromonas hydrophila by agricultural waste extracts and assessment of aerolysin inhibitors using in silico approach" is a collection of multiple approaches taken against Aeromonas hydrophila, however, multiple issues were detected.

First, the language in the entirety of the article requires extensive reviewing since, at times, it is very hard to understand what it is being said.

Second, while the introduction focus a number of topics that will be discussed in the article, mainly biofilm, quorum sensing and aerolysin, there was very little effort in integrating the multiple themes in a comprehensible line of reasoning.

Methodologically, there are also multiple issues with the approaches taken that should be addressed:

- Regarding the extracts used, there isn't a description of their chemical composition or major components, making the interpretation of much of the data presented much less clear, possibly even pointless, and is an issue especially for the computational studies.

- The biofilm study was based only in SEM images, hence additional controls would be necessary, at least containing a known biofilm inhibitor. However, multiple approaches should be taken in order to justify the observed effect as biofilm inhibition. The data presented clearly is not enough to justify such affirmation, unless strongly based in literature. Furthermore, the authors make the statement that the biofilm reduction happens by blocking the Quorum Sensing, without providing experimental data to backup or even strong evidence in the literature presented.

- The authors presented the FT-IR analysis of Aeromonas hydrophila biomass but it was not clear to me what was the initial purpose of this analysis or its major conclusions.

- The computational approach is described as an assessment of aerolysin inhibitors, which is a statement that is lacking validation in multiple points. Mainly, if the strategy was to study the inhibition of the formation of aerolysin aggregates, the approaches taken should take into consideration multiple units of the protein, and a model of its aggregation studied, then, the ability of ligands to disturb the aggregation model or even prevent it could be studied.

In summary, most of the conclusions presented lack validity, experimental and literature support, hence the work presented requires extensive reviewing.

Author Response

Comments and Suggestions for Authors

Reviewer 5:

The work presented entitled “Evaluating biofilm inhibitory potential in fish pathogen, Aeromonas hydrophila by agricultural waste extracts and assessment of aerolysin inhibitors using in silico approach" is a collection of multiple approaches taken against Aeromonas hydrophila, however, multiple issues were detected.

First, the language in the entirety of the article requires extensive reviewing since, at times, it is very hard to understand what it is being said.

Author’s response: As per the reviewer’s suggestion, the language of the revised manuscript was extensively reviewed by an English expert to improve the legibility. The certificate for the language proof is given below.

Second, while the introduction focus a number of topics that will be discussed in the article, mainly biofilm, quorum sensing and aerolysin, there was very little effort in integrating the multiple themes in a comprehensible line of reasoning.

Author’s response: We thank the reviewer for this valuable suggestion. This study mainly focusing on the development of the drug from the agricultural wastes to treat against fish pathogen, Aeromonas hydrophila. Microbial colonies known as biofilms attach to surfaces and are encased in an extracellular matrix that they have developed on their own. Due of their great antimicrobial resistance, these communities are challenging to eliminate. Interfering with quorum sensing (QS) systems, which are important in controlling biofilm development and pathogenicity in many bacterial species, is one strategy that reduces the production of biofilms. To support the inhibitory potential of the extracts, antibiofilm activity has been analysed by SEM. The primary strategy for controlling the antibiofilm capacity is to interfere with quorum sensing signals in order to prevent cell-cell communication. And inhibition of aerolysin by metabolites has potential implications for the development of new therapies for infections caused by A. hydrophila. In these aspects, the introduction section describes about biofilm, quorum sensing and aerolysin in the separate paragraph in the revised manuscript.

Methodologically, there are also multiple issues with the approaches taken that should be addressed:

- Regarding the extracts used, there isn't a description of their chemical composition or major components, making the interpretation of much of the data presented much less clear, possibly even pointless, and is an issue especially for the computational studies.

Author’s response: We thank the reviewer for valuable suggestions, the major bioactive compounds present in the solvent extracts of both GNS and BGP were already published and the citations have been in the revised manuscript. Page number 16; line number 240-254: Page number 17; line number 291-299.

Based on our previous results, both GNS and BGP solvent extracts possess phenols and tannins [56-57]. These primary bioactive compounds are the major cause which exhibits better antioxidant and antibacterial properties [58-59]. The metabolites from the polar solvents also tend to diffuse the fatty acid composition of the rigid layer of A. hydrophila [60]. The metabolites of plant extracts may act on reducing the colonization of body surfaces and different epithelial layers, certain inorganic and organic molecules. Other micro and macronutrients necessary for cell growth also promote cell adhesion [61]. After 48 h treatment with the extracts of both GNS and BGP, A. hydrophila cells were shrunken. They underwent splitting due to metabolites like palmitic acid, methyl linoleate, H-Pyran-4-one-2,3 dihydro-3,5 dihydroxy-6-methyl and 2-Hexyldecanoic acid [62-65]. These metabolites adhered to the lipopolysaccharides of the cell membrane, thus altering the bacterial cell morphology [66]. The metabolites from the extracts may inhibit nutrient availability that paved the way for bacterial cell growth [67]. The formation of the matrix by bacterial cells was separated due to inhibiting quorum sensing signals from one cell to another [39, 68].

Essentially, Polyphenols and bacteria interact in a non-specific manner, relying on the hydrogen group and hydrophobic effects that may have a significant influence owing to lipophilic interactions and the creation of covalent bonds [95]. Phenolic compounds present in the extracts may directly interact with the bacterial cell membrane, which causes intracellular leakage and ROS generation [96]. FT-IR analysis of A. hydrophila biomass can reveal important information about the bacterium's chemical composition, such as the presence of proteins, lipids, and fatty acids [97]. This knowledge can help us understand the structure and function of the bacterial cell, as well as create ways to prevent or treat A. hydrophila infections.

 - The biofilm study was based only in SEM images, hence additional controls would be necessary, at least containing a known biofilm inhibitor. However, multiple approaches should be taken in order to justify the observed effect as biofilm inhibition. The data presented clearly is not enough to justify such affirmation, unless strongly based in literature. Furthermore, the authors make the statement that the biofilm reduction happens by blocking the Quorum Sensing, without providing experimental data to backup or even strong evidence in the literature presented. 

Author’s response: Based on the reviewer’s comments, the SEM images of streptomycin treatment (positive control) were also included along with the recently added discussions for the biofilm reduction mechanism by blocking the quorum sensing signals in the revised manuscript that support our research findings in the present study. And also, the exact concentration of the extracts determined for Minimum inhibitory concentration against A. hydrophila have been given in Table 3.

Page number 16; line number 240-271.

- The authors presented the FT-IR analysis of Aeromonas hydrophila biomass but it was not clear to me what was the initial purpose of this analysis or its major conclusions.

Author’s response: Based on the reviewer’s critic, The purpose of using FT-IR technique in this study and its discussion were given page number 16-17; line number 277-289: Page number 19; line number 410-413 of the revised manuscript.

- The computational approach is described as an assessment of aerolysin inhibitors, which is a statement that is lacking validation in multiple points. Mainly, if the strategy was to study the inhibition of the formation of aerolysin aggregates, the approaches taken should take into consideration multiple units of the protein, and a model of its aggregation studied, then, the ability of ligands to disturb the aggregation model or even prevent it could be studied.

Author’s response: As per the reviewer’s comment, the inhibition of the aerolysin aggregation by the ligands and bioactive compounds from the extracts have been included in the revised manuscript with suitable references. Page number 17; line number 320-329. References cited were given in the bibliographic section as 109-111.

Ligands are compounds that can control the activity of a protein or enzyme by binding to specific sites on the target protein or enzyme. In the case of aerolysin, ligands can be employed to prevent the production of toxin aggregates, which can injure host cells and tissues [109]. Ligands can bind to different sites of aerolysin, such as hydrophobic re-gions on the toxin's surface, particular spots on the pore-forming domain, and other sections of the molecule [110]. Flavonoids and polyphenols have been demonstrated to suppress the production of aerolysin aggregates. These bioactive compounds can attach to particular sites on the toxin and prevent it from building huge complexes that can damage host cells. This kind of in vitro approach to be beneficial in decreasing aerolysin toxicity [111].

In summary, most of the conclusions presented lack validity, experimental and literature support, hence the work presented requires extensive reviewing.

Author’s response: We thank the reviewer for the valuable suggestions, the concluding remarks of the each experiment and the respective results were discussed elaborately and sentences were modified for the reader’s convenient in the revised manuscript. The manuscript was reviewed extensively and substantial changes were made. We once again thank the reviewer for these suggestions and helped to improve the scientific quality of the manuscript.

Page number 16; line number 240-271: Page number 16-17; line number 277-289: Page number 17; line number 291-299, 304-307, 320-329: Page number 18; line number 348-350 & 366-368. Page number 20; line number 459-461 & 464-466.

Round 2

Reviewer 1 Report

The authors have addressed my comments.

Reviewer 5 Report

First of all, thank you to the authors for taking the time to address all of my concerns.

The idea presented in the document is highly relevant and commendable, as the reuse of subproducts is an essential approach for sustainability, especially when it can help combat important pathogens in aquaculture.

The updated document has certainly improved, and the language is now more comprehensive. However, after careful consideration and literature review, there are still some core scientific questions that remain.

Regarding the various topics discussed in the article, the integration of information is much clearer and follows a logical line of reasoning. However, the connection between the information presented and the experimental methodologies applied is still unclear. For instance, while it is known that biofilm production can be modulated by quorum sensing (QS) systems, none of the methodologies presented in the article can provide a clear assessment of whether the QS system is being affected. As the authors themselves note in their discussion, the gene expression system for QS is well-known, as is its relevance to exopolysaccharide production. Therefore, suitable approaches to determine whether there is an impact on these pathways would be to study whether the extracts are altering gene expression at the level of quorum sensing or measuring the production of exopolysaccharides and determining if there is any variation with exposure to the extracts in a concentration-dependent manner, particularly at sub-inhibitory concentrations, since bacterial death can be a confounding factor.

However, the study presented employs scanning electron microscopy (SEM), which, as noted by the authors, shows compromised bacteria with multiple structural modifications that are not dissimilar to the control condition. Therefore, it is not possible to establish a connection between the lack of biofilm and QS inhibition, as the absence of biofilm could also result from bacterial death, as in the control condition.

Additionally, regarding the FT-IR analysis, the authors state that "the intensity peak at 1120-1160 cm-1 indicates the presence of polysaccharides in both control and treated groups due to bacterial biofilm formation" contradicting their previous discussion. Furthermore, the analysis of the results shows no discriminatory capacity between the control and treated groups, making it impossible to draw any significant conclusions from this analysis.

Regarding the computational approach taken, multiple methodological errors can be observed, as I noted before, and most of the literature does not support the methodologies applied. For example, Reference 109, which supposedly reports on the effectiveness of ligands, is actually referring to the effectiveness of proteins, not small molecules, as is the case with the current report. Furthermore, a computational methodology to address the issue of the formation of toxic aerolysin must account for its complex structural features, as described in Reference 110. The structural model used in the current report was built by homology, but did not compare or use any of the multiple crystallographic structures available for proaerolysin of Aeromonas hydrophila or of the complete heptamer.

Unfortunately, these are just a few examples of the methodological inadequacies that undermine the conclusions presented in the document. Therefore, I would recommend that the authors reassess their methodologies, ensure that their conclusions are well-supported, and resubmit a new document in the future.